# Life Cycle Assessment Model of Plastic Products: Comparing Environmental Impacts for Different Scenarios in the Production Stage

**DOI:** 10.3390/polym13050777

**Published:** 2021-03-03

**Authors:** Viktoria Mannheim

**Affiliations:** Higher Education Industrial Cooperation Center, University of Miskolc, 3515 Miskolc-Egyetemváros, Hungary; mannheim@uni-miskolc.hu

**Keywords:** mixed-plastic product, polypropylene product, production life cycle stage, life cycle assessment

## Abstract

This paper assesses the environmental loads of polypropylene and PP-PE-PET mixed-plastic products throughout the products’ life cycle in the production stage, with particular focus on the looping method. A life cycle model of homogeneous and mixed-plastic products has been developed from the raw material extraction and production phase through its transport with the help of the life cycle assessment method. To find the answers to the questions posed, different impacts were analyzed by the GaBi 9.5 software. The analysis lasted from the beginning of the production process to the end. The aim of this research was to determine the energy and material resources used, the emissions produced, and the environmental impact indicators involved. This article examines three scenarios in the production stage, based on the usage of plastic scrap and process water: (1) plastic scrap and wastewater are recirculated with looping method; (2) plastic scrap goes through an incineration process and wastewater is treated in a municipal wastewater treatment plant; (3) plastic scrap is sent to a municipal landfill and wastewater is treated. This article tries to answer three questions: (1) how can we optimize the production stage? (2) Which materials and streams are recyclable in the design of the life cycle assessment? (3) What is the relationship between the environmental impacts of homogeneous and mixed-plastic products? The results of this research can be used to develop injection-molding processes with lower environmental impacts and lower releases of emissions.

## 1. Introduction

Life cycle assessment (LCA) is a methodology proposed by Bicalho et al. that is used to evaluate the environmental impacts of products from the production of raw materials to the products’ end of life [1]. According to the scientific works of Jensen et al. [2], life cycle analyses flows were initially determined for the depletion of material and energy resources. Life cycle assessment is one of the most common and reliable methods by Klöpffer and Grahl [3]. Life cycle methodologies can be applied to several aspects of production. To achieve process optimization, life cycle assessment can be successfully used to analyze the environmental impact of different stages of a product’s life cycle [4,5]. The LCA process begins with the determination of goals and system boundaries. In cases where boundaries are well defined, the following step is life cycle inventory analysis (LCI) and then life cycle impact assessment (LCIA). Finnveden et al. [6,7,8] review the differences in life cycle assessment methods. The variety of developed databases and software programs provide us the opportunity to devise environmental impact reduction solutions at different life cycle stages, as demonstrated by Bach et al. [9].

The use of plastic products in the technological sphere has, in recent years, received increased attention. The use of injection-molding processes is widespread in the manufacture of plastic products. Due to the importance of these processes, scientific research is increasingly concerned with the life cycle analysis of molded products [10,11]. Life cycle assessment is one of the most appropriate methods for analyzing the environmental impact of a polymer product.

In the context of circular economy and sustainability, it is important to evaluate the life cycle of plastic products. In accordance with the European Environment Agency, the practical solutions aimed at establishing a circular economy include extending the life cycle of products [12]. Many research studies have argued that the circular economy promotes minimizing material and energy flows and reducing waste [13]. Focusing on the safety of the circular economy can take LCA to a new level and set additional targets for reduced environmental impact. The idea of a full life cycle in this research topic has been raised by Civancik-Uslu et al. [14,15]. According to the conclusions of Erdélyi et al. [16], in accordance with the life cycle assessment approach, the carbon footprint associated with the phases of transport and waste management processes should also be specified.

In assent with the scientific results of Labuschagne et al. [17], in the innovative technological developments based on life cycle assessments, the manufacturing stage must be considered especially. At the production stage, we need to consider that plastic waste can be treated via disposal, incineration, or recycling processes. In assent with LCA studies [18,19] and European Food Safety Authority (EFSA) [20], recycling plastic waste is generally environmentally preferred. The quality of the recycled polymer product mainly depends on the physicochemical properties of the polymer, as well as the processing conditions and the purity of the input waste. Several studies have attempted to address economic and ecological issues by limiting the widespread use of chemical recycling processes [21,22]. Life cycle analysis with looping method in the manufacturing stage promotes sustainable production by maintaining the value of products [23]. In the production phase, the use of renewable raw materials has increased in recent years and waste generation has decreased [24]. Plastic scrap looping has the potential to become a new and effective strategy in the production stage within the new framework defined by sustainable production and the circular economy. In agreement with the scientific studies of Villares et al. [25], the use of life cycle assessment at early planning stages gives neither final nor accurate results but can have a great impact on the environmental performance, especially when comparing scenarios. the application of life cycle assessment in the early planning stages gives neither definitive nor accurate results, but can have a major impact on environmental performance, especially when comparing different scenarios. Grosso et al. [26] introduced beneficial solutions involving the use of recycled scrap instead of virgin material. A life cycle analysis of plastic products with the use of scrap and process water looping has not been previously performed within the European Union. The principle of reduction aims at minimizing the use of raw materials, energy, and waste during the production phase, while the principle of re-use aims at the proper re-use of products (Ghisellini et al. [27]).

The aim of this research was to analyze the environmental burdens of a polypropylene product and a mixed-plastic product throughout the product’s production phase, focusing on the injection-molding stage. The main goal was to investigate information on the impact categories and resources related to the injection-molding of polymer products. In this research, three different solutions were examined: (1) the use of recirculated plastic waste and water looping; (2) not using the looping, where plastic scrap is incinerated and wastewater is treated; and (3) without looping, where scrap is landfilled and wastewater is treated.

The first goal of this research work was to compare the environmental impact of the different plastic products’ manufacturing phases. The second objective was to propose an injection-molding technology that offers an enviro-friendly technological solution. The first section of this paper explains the methodology, including our goal, different examined scenarios, the determination of the functional unit and the system boundaries, the allocation method, and the applied software. The first part of this article explains the methodology and the applied methods, including the functional unit and system boundaries, the different scenarios, the allocation method, and the life cycle assessment software. The next part describes the life cycle inventory and evaluates the life cycle impact assessment.

This paper introduces the calculated environmental impact categories, the material and energy resources, and the emission values. The main section provides the obtained research results for three scenarios. The last section presents the conclusions of the research work. The results serve the design of manufacturing processes with lower environmental impact and the improvement of the environmental performance of the injection-molding process. This work is relevant because the reduction in resources and environmental impacts is authoritative in a life cycle analysis.

## 2. Materials and Methods 

### 2.1. Methodology and Scenarios

The life cycle assessment methodology was applied in accordance with the recommendations of ISO 14040 and 14044 standards [28,29]. The ISO 14040 standard defines the LCA methodology by explaining different life cycle phases. The goal phase defines the context of the research study. The life cycle inventory phase identifies the inputs that are inputs to the system. The life cycle impact assessment phase defines the environmental impact categories. In the interpretation phase, the information from the results are evaluated [5,28,29]. This research study comprises the life cycle inventory phase, the life cycle impact assessment phase, and the interpretation of the results. This approach enables the analysis of the environmental impacts associated with the manufacturing stage in the life cycle of the plastic products (from different plastic granules), from the extraction of raw materials for their injection-molding until the plastic scrap becomes plastic waste. The homogenous polypropylene and PP-PE-PET granules are produced in the European Union and processed in the local injection-molding plant of polymer products. The granules are molded and the plastic scrap is managed as plastic waste in an incineration plant or on a landfill. 

This paper analyzes three scenarios that define the life cycle of the different plastic products. Different methodologies have been developed for the distribution of material and energy consumption and the resulting emissions and waste. Two parameters (plastic scrap and process water) and two plastic products from different polymer granules were analyzed in terms of their response to possible changes in the manufacturing stage to identify which of the different examined scenarios is preferable from an environmental view.

In the first scenario, the injection-molding process is examined with looping-method. Here, plastic waste is recycled, and the cooling water is managed in 50% as wastewater. The second and third scenarios determine the manufacturing process without looping. In the second scenario, plastic waste is incinerated, and wastewater is handled in a municipal wastewater treatment plant. The third scenario determines the resources and environmental impacts when plastic scrap is landfilled, and wastewater goes through wastewater treatment. The effects of replacement and recycling were not calculated. The input‒output data for the production stage were calculated considering the reference flows.

This research provides new information regarding the objective environmental impacts associated with the production of different polymer products in the European Union by comparing different scenarios for the injection-molding process with a looping method and without looping using plastic waste and wastewater treatment technologies.

### 2.2. System Boundaries, Functional Unit and Allocation

This research examines the life cycle of the manufacturing stage, taking into account the stage of extraction of raw materials needed for injection-molding. The manufacturing process was assigned as a function of the mass of the molded polymer. The system boundaries were developed from the beginning to the end. Datasets were linked with injection-molding process data to create life cycle inventories for the examined plastic products. Auxiliary systems included transporting materials for production, obtaining electric power from a Hungarian energy mix, and diesel oil for the transportation of the raw materials. Plastic products have a 20-year lifetime and the analyzed amount was molded in 1 h. 

Considering the effects of the life cycle of products in the manufacturing phase, the functional unit is defined as the distribution of 28 kg of product output. In the production stage all materials and energy that were used as well as all emissions that were produced are related to the plastic product of the injection-molding process. In addition to the main polymer products, this process produces plastic waste and wastewater. 

The allocation hierarchy we used was that suggested by ISO 14044 [29]. For the transport of refinery products (diesel oils), the emission allocation was by mass. The energy demand was taken into account as a function of the input energies. Recycling of plastic waste would reduce the environmental impact by following the distribution method and ignoring the collection and transport stages. Therefore, we can optimize the injection-molding process with a lower environmental load.

### 2.3. LCA Software

The aim was to determine and quantify the resources, emissions, and environmental impacts for the different polymer products in the production stage with professional and extension dataset. The analysis of the tested system was performed by using GaBi 9.5 thinkstep software (Sphera Solutions Ltd., Stuttgart, Germany). Normalized and weighted values for the different scenarios were determined by the Higher Education Industrial Cooperation Centre (HEICC) of the University of Miskolc. The applied LCA software provided valuable resources to support the consistent modelling of the production life cycle. The results from the LCA software highlight the estimated environmental performance in terms of various aspects, such as carbon footprint, resource and energy consumption, and various environmental impacts [30].

### 2.4. Life Cycle Inventory Methodology and Life Cycle Impact Assessment Method

In this research analysis, the environmental effects associated with the production life cycle of plastic products are considered. Cradle-to-gate data for polymer products are provided to illustrate the contribution of the converting process to the life cycle inventory results to produce injection-molded products. The quality of the life cycle inventory method is directly related to the accuracy of the data. The applied life cycle inventory method includes the input‒output material flows and energy requirements for the unit process. For the modelling of injection-molded product systems, we used product-specific input data. This methodology distributes energy demand and emissions among molded products in addition to mass allocation. The dataset for polymer granules includes annual averages. The life cycle inventory is primarily in line with industry data on internationally prevalent manufacturing processes.

Accounted resource inputs include material and energy use, while process production includes plastic products and emissions to soil, air and water. In the manufacturing stage, each dataset includes incoming transportation. The injection-molding process was also examined as a looped system by accounting for all resource inputs and process outputs. I used allocation to divide the environmental load of process water and plastic waste. The amount of energy used to heat, cool, and illuminate the injection-molding space is not included in the LCI system boundaries. The life cycle inventory does not include the following components: investment vehicles, miscellaneous materials and additives. The applied methodology is in line with the life cycle inventory described in ISO 14040: 2006 [28].

The aim of the life cycle assessment is to find the optimal process conditions, which will lead to a reduction in the consumption of natural resources [31]. LCA phases include goal and scope definition, life cycle inventory, environmental impact assessment, and interpretation of the results [28,31]. The life cycle impact assessment method can be used to determine the relative risk of emissions from the system under study to humans or the environment. The Energy and Climate Policy Framework for 2030 established ambitious commitments to reduce greenhouse gas emissions by at least 40% by 2030 [31,32]. There are many aspects of ISO 14040 standards for the design and implementation of life cycle assessment [28,33]. The life cycle impact assessment phase aims to investigate the possible environmental impacts in the studied system [34].

During the life cycle impact assessment, the reference system was the total inputs and outputs for the European Union. In Europe, the focus is on the characterization methods, I applied the CML (the Institute of Environmental Sciences) method. In this method, the impact categories have been developed by the Centre for Environmental Science at Leiden University [35,36,37]. The normalization and weighting methods were the same for all analyses performed. The applied normalization reference represented the environmental impacts of 28 European Union countries. The weighting method for the LCIA survey in 2012 was CML 2016 for Europe. Eight environmental impacts—global warming, eutrophication, acidification, photochemical ozone creation, human toxicity, abiotic depletion (fossil and elements), and marine aquatic ecotoxicity—were used for this research analysis. The potential value of global warming is for 100 years, except for biogenic carbon. Sulphur, nitrogen oxides, and phosphorous compounds are directly related to eutrophication and acidification potentials. The formation of tropospheric ozone was considered in terms of the photochemical ozone creation potential. Human toxicity potential describes the effects of toxic substances. Abiotic resource depletion is one of the most debated impact categories. Guinée et al. [38,39,40] based the characterization model of abiotic resource depletion on physical data on reserves and annual deaccumulation. Marine aquatic ecotoxicity refers to the effects of toxic substances released into marine aquatic ecosystems.

In this paper, the functional unit value was 28 kg of plastic product and different environmental loads were investigated for the life cycle of the product. A standard unit of output was used as the basis for determining the life cycle resource requirements and environmental emissions of different plastic products of 28 kg. By the assessment of the production stage of the investigated plastic products and its optimal approaches, the life cycle assessment results for different products were compared by quantifying the environmental impacts, resources, and emissions. The environmental loads were calculated for all scenarios and summarized to compare the environmental impacts of the polymer products.

## 3. Results and Discussion

### 3.1. Production Stage Setup Process

The life cycle of the plastic products can be divided into different stages based on the environmental product declaration (EPD) modules [41]. As shown in Figure 1, points of intervention for material and energy efficiency strategies can occur throughout a product’s life cycle by EPD technological modules. In the manufacturing stage (A1–A4), numerous factors and environmental loads must be considered. This phase encompasses the supply of raw materials (plastic granules, compressed air, and process water), energy supply, the transport of raw materials, and the injection-molding process. The production stage of the examined plastic products is based on an injection-molding process under actual operating conditions. At the production stage, I worked with the latest data from the GaBi 2020 database. This stage involved injection-molding the plastic granules as well as shipping the plastic product to be used. During the life cycle assessment of the injection-molding process, the background of the raw and auxiliary materials was also considered. These data are valid for the period 2019–2022. The raw materials used for the injection-molding process are homogeneous polypropylene and mixed-plastic granules, compressed air, and process water, where 30 kg of granules form 28 kg of plastic product in 1 h. The polypropylene granules are a polypropylene technology mix. The mixed-plastic granules are the author’s own PP-PE-PET mix (in an equal weight distribution) from EU-28. For electricity, the Hungarian energy mix was introduced in the latest statistical database.

In this research analysis, the life cycle of a polypropylene product and a mixed-plastic product were examined from the raw material extraction to the end of the production stage with transports. I designed with a maximum product loss of 7% in the manufacturing phase. The examined scenarios in the production stage are as follows:

Scenario 1: The plastic waste and the wastewater are looped.

Scenario 2: The plastic waste is incinerated and the wastewater is treated (without the looping method).

Scenario 3: The plastic waste is landfilled and the wastewater is emitted to a municipal wastewater treatment plant (without the looping method).

First, I determined the mass and energy values of the input-output parameters of the injection-molding process of pure polypropylene and plastic mix products, such as process water, plastic granules, electricity, and compressed air. The waste flows (plastic scrap and wastewater) generated in the production stage were determined. 

I applied dummy processes in the software to design the production. I connected a flow output to the same process as the input. According to the first scenario, I circulated the water flow in a 50% closed loop, and the other part of the water flow was handled as wastewater. The life cycle assessment processes and plans were set up for the production life cycle stage in the applied GaBi 9.5 software.

### 3.2. Environmental Impact Results for the Different Scenarios

The environmental impacts of water consumption on the product systems were assessed with the LCIA methods developed by Pfister et al. [42] as well as the method proposed by Frischknecht et al. [43,44]. In the second scenario, the product loss was treated as plastic scrap with incineration, and in the third scenario the plastic scrap was landfilled. The transport distance was 100 km with utilization of 85%, taking into account road transport in the European Union. The electricity and refinery products were modelled in harmony with the individual country-specific situation. Table 1 describes the examined environmental impacts in this research analysis.

Figure 2 and Figure 3 present the normalized and weighted values for eight environmental impact categories in nanograms under the injection-molding conditions of Scenario 1. These figures clearly show that the marine aquatic ecotoxicity potential (525–586 kg) and abiotic depletion for fossil fuels (364–383 kg) are higher compared to other environmental impact categories. The values of elements abiotic depletion are negligible for both plastic products (0.49–0.51 kg).

Figure 4 and Figure 5 show the normalized and weighted values for the environmental impact categories for Scenario 2. These figures clearly show that the abiotic depletion for fossil fuels (379–400 kg) and the global warming potential for 100 years (111–130 kg) are higher than other environmental impact categories. The values of abiotic depletion for elements are negligible for both plastic products (0.51–0.54 kg).

Figure 6 and Figure 7 illustrate the normalized and weighted values for environmental impact categories for Scenario 3. The figures show that the marine aquatic ecotoxicity potential (616–623 kg) and abiotic depletion for fossil fuels (388–409 kg) are higher. The values of abiotic depletion for elements are very low (0.52–0.55 kg).

### 3.3. Comprising of Environmental Impacts for Different Scenarios

In this research study, eight environmental impacts and the value difference associated with the plastic products for the three different scenarios are analyzed. Table 2, Table 3 and Table 4 describe the examined environmental impacts in the research analysis. The highest environmental load comes from the granule production itself. The acidification potential (AP), photochemical ozone creation potential (POCP), global warming potential (GWP), and eutrophication potential (EP) impact categories have a greater difference on the life cycle assessment results of the examined products. In summary, we can say is that the values of all the effect categories are greater for the mixed-plastic product (the exception for Scenarios 2 and 3 is the value of marine aquatic ecotoxicity potential (MAETP).

Table 5 and Table 6 illustrate the values of environmental impacts associated with the production life cycle of a 28 kg plastic product for the three examined scenarios. In summary, the values of all the impact categories are greater for Scenario 3. The exception is the value of global warming potential. There are basically minimal differences in the GWP value, but they are slightly higher for Scenario 2.

However, the value of all the impact categories is higher for the mixed product except for marine aquatic ecotoxicity (MAETP). The value for MAETP is slightly lower for the mixed-plastic product in Scenarios 2 and 3. If ecotoxicity is not assessed, then the highest values are for global warming potential (GWP) and abiotic depletion for fossil fuels (ADPF); these environmental impact categories have been illustrated in nanograms. Figure 8 shows the global warming potential of polypropylene and mixed-plastic products for the three scenarios in the production stage with transport. Figure 9 presents the abiotic depletion for the fossil fuels of both products for each scenario in the production stage with transport. In the case of GWP, we see a difference of 3–4%; in the case of ADPF, there is a difference of 1–1.5% for each scenario.

In both examined production processes, except for global warming potential the values of all the impact categories are greater for Scenario 3. In this case, the plastic scrap is traditionally deposited as plastic waste. The global warming potential value is greater for Scenario 2, where the plastic waste is incinerated. For all the examined scenarios except for marine aquatic ecotoxicity, the environmental impacts are greater for the mixed-plastic product.

### 3.4. Material and Energy Resources for the Different Scenarios

In the second scenario, the end-of-life of the plastic scrap was modelled with the waste incineration of plastics in a municipal waste incineration plant in the European Union. In the third scenario, the end-of-life of the plastic scrap was modelled with the landfilling of plastics in a municipal landfill plant in the European Union. Many studies summarize information for the waste management processes of plastic waste with a comparison between the different technologies available [45,46,47,48]. The conventional incineration process and landfilling can be compared on the basis of their environmental impacts and energy efficiency. Waste-to-energy (WTE) incineration plants are inextricably linked to the circular economy system, social harmony, environmental outcomes, risk assessment, and energy transformation [49]. Taşkin et al. [50] evaluated three different municipal solid waste (MSW) management strategies with LCA method in terms of environment and energy. Dastjerdi et al. [51] adopted the LCA method to study the potential of WTE technologies in the areas of energy recovery and greenhouse gas emissions. In the energy aspect, it is important to transform energy from residual MSW and assess the potential energy recovery from waste in relation to the circular economy strategy [49,52]. WTE incineration plant site selection can be considered as a multi-criteria decision-making problem [49].

For the transportation of plastic waste, the transport distance was 100 km by road within the European Union into account. The raw materials and energy streams used determine energy consumption and environmental impacts, so they can affect the production phase and life cycle of plastic products. Residual steam and electrical energy must be reused in a specially designed plant.

Material resources, energy resources, and emissions to freshwater and air are larger than other flows; therefore, these parameters were illustrated. The percentage of these parameters was 48–49% for material resources and 51% for emissions to freshwater. The values of the other emissions examined (deposited goods, emissions to seawater, and emissions to agricultural and industrial soil) were small for the three scenarios examined. The percentages of emissions to sea water were 0.09–0.18%. Therefore, these flows are not shown in the figures.

Figure 10 and Figure 11 present the values of material and energy resources for the three examined scenarios in the production stage. Figure 12 and Figure 13 show the values of emissions to freshwater and air for the three examined scenarios.

The highest resources and emissions to freshwater for both examined products were observed in Scenario 3. The highest emissions to air for the plastic products were observed in Scenario 2. The applied looping method in Scenario 1 reduced the material resources by 4.4–4.6% and reduced the energy resources used by 6%. The looping method reduced the emissions to fresh water by 4.1–4.2%. In relation to Figure 13, we can determine that the largest change was observed in the air emissions. The looping method when used in the production stage can reduce the emission to air values by 6.6% over the life cycle of the tested products.

The use of pure polypropylene granules causes 6–7% higher environmental loads in terms of material resources and freshwater emissions for all three scenarios. The use of mixed-plastic granules causes 4% higher environmental loads in terms of energy resources and freshwater emissions for all three scenarios. The use of mixed granules causes 13% higher environmental loads in terms of air emissions for all scenarios.

## 4. Conclusions

This research study examined the manufacturing life cycle of a polypropylene product and a mixed-plastic product, from the extraction of raw material to the injection-molding and plastic scrap treatment processes, and attempted to determine its impact on the environment. The life cycle assessment included a scenario analysis in the manufacturing phase. Regarding the different scenarios, two parameters (plastic waste and cooling water) and two products (pure polypropylene and PP-PE-PET mixed plastic product) were examined to identify which scenario is more optimal. In the first scenario, the injection-molding process was determined with plastic scrap and process water looping. In this case, plastic scrap was recycled and 50% of the water from the cooling process was managed as wastewater in a wastewater treatment plant. The second and third scenarios determined the injection-molding process without scrap and water looping. In the second scenario, plastic scrap went through incineration and wastewater was treated. In the third scenario, plastic waste was deposited, and wastewater went through wastewater treatment. Thus, the applied life cycle model was completed with wastewater treatments, plastic waste landfilling, and waste incineration.

By quantifying the environmental impact categories of all scenarios, the CML analysis method was applied by the GaBi 9.5 think step software. The normalization and weighting methods were the same for all scenarios. Material and energy input-output flow from injection-molding had a significant impact on the life cycle of the product. The functional unit was defined as the distribution of 28 kg of polypropylene and plastic mix product output for all stages. The lowest environmental effects of the products were recorded in Scenario 1. In this scenario, I performed the recycling of materials (cooling water and plastic loss) using the looping method.

According to the research results of the life cycle assessment, the looping method in Scenario 1 reduced the values of examined impact categories by 6–7%. Examining the three scenarios in parallel, it can be concluded that the largest value differences were found in POCP (20–21%), AP (17–23%), GWP (14–15%), and EP (14–25%). The value of all the impact categories was higher for the mixed-plastic product, except marine aquatic ecotoxicity. The values of MAETP were lower for the mixed product in Scenarios 2 and 3. If we compare Scenarios 2 and 3 for the manufacturing phase of the pure polypropylene product, it can be said that the value of marine aquatic ecotoxicity (MAETP) was 6.1% and the value of acidification (AP) was 1.44% higher in the case of plastic waste landfilling. Comparing Scenarios 2 and 3 for the manufacturing phase of the mixed-plastic product, the value of MAETP was 6.42% and the value of AP was 1.34% higher in the case of landfilling. For the other analyzed impact categories, we obtained similar values for the pure polypropylene and the mixed product during disposal and incineration.

The highest resources and emissions to fresh water for the plastic products were observed in Scenario 3. The highest emissions to air were observed in Scenario 2. The applied looping method reduced the material resources by 4.4–4.6% and the energy resources by 6%. The looping method reduced the emissions to fresh water by 4.1–4.2%. It could reduce the emission to air values by 6.6% over the life cycle of the examined plastic products. In summary, it can be stated that the largest change was observed in air emissions, and the use of plastic mix granules caused 4% higher environmental loads in terms of energy resources and freshwater emissions. At the same time, the use of mix granules caused 13% higher environmental loads in terms of air emissions for all the scenarios.

From research to policymaking, the major challenge is to find the aspects that have the greatest effect on environmental impacts, thereby fostering eco-innovation [53]. For emerging technologies and products, identifying environmental hotspots and informing decision makers is a crucial starting point for sustainable product and process development [53,54]. Life cycle assessment follows a four-phase approach (goal and scope, life cycle inventory, impact assessment and interpretation) according to the ISO 14040 series of standards [28].

To achieve sustainable production, it is essential to analyze the life cycle of the product in the manufacturing phase. Vuarnoz et al. [55] presented an energy management procedure that optimized the emissions during the first life cycle phase. However, I did not find any studies that looked at a life cycle model with the looping method for injection-molding process with a comparison of different plastic products. Life cycle assessment approaches could help to facilitate the comparison of different alternatives for individual materials and products for a global rating. Cradle-to-gate LCA refers to material development using a cradle-to-gate approach, and the results of this LCA could be used as an input in further LCAs [56].

These results can be optimized injection-molding processes with favorable impacts on the environment. Furthermore, my results can be applied to further research on the injection-molding processes of other polymer products. The values of environmental impact categories are high in the production stage. However, in the first scenario I demonstrated for two examined plastic products that it may be possible to decrease the impacts on the environment if the production process was carried out more sustainably. The environmental loads can be decreased by applying the LCA looping method. These results support the product-oriented environmental management of plastic products. The most economical technologies are more environmentally friendly and energy efficient and can improve the economic efficiency of companies The results of the life cycle analyses presented in this research are expected to contribute to a better understanding of the life cycle of plastic products in the European Union.

## Figures and Tables

**Figure 1 polymers-13-00777-f001:**
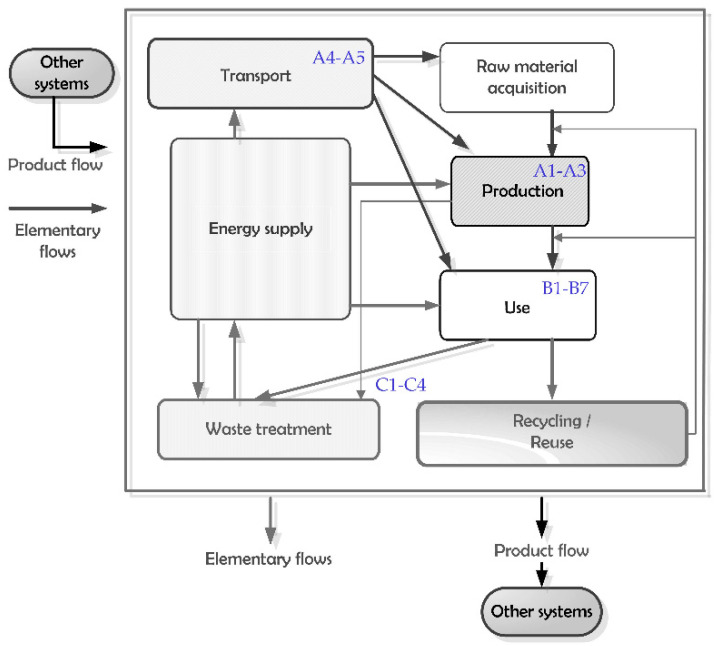
Product life cycle by environmental product declaration modules.

**Figure 2 polymers-13-00777-f002:**
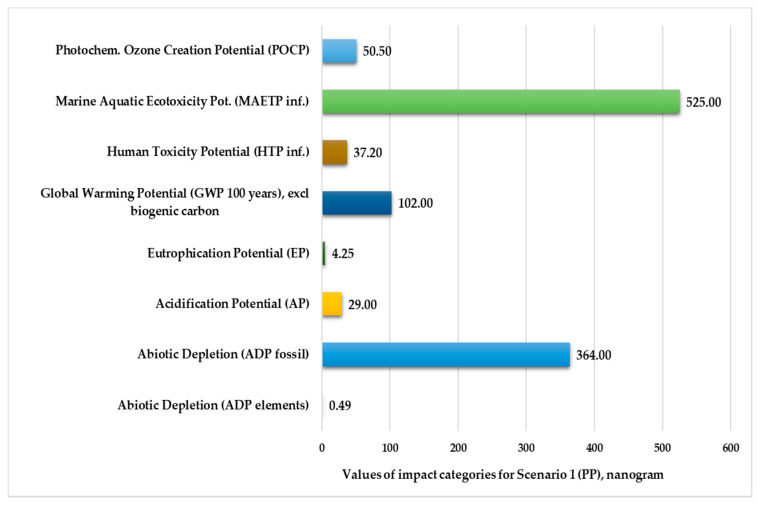
Environmental impact categories for Scenario 1 in the production stage with transport (functional unit: 28 kg polypropylene product. Normalization reference: CML 2016, EU 25+3, year 2000, excl. biogenic carbon. Weighting method: thinkstep LCIA Survey 2012, Europe, CML 2016, excl. biogenic carbon).

**Figure 3 polymers-13-00777-f003:**
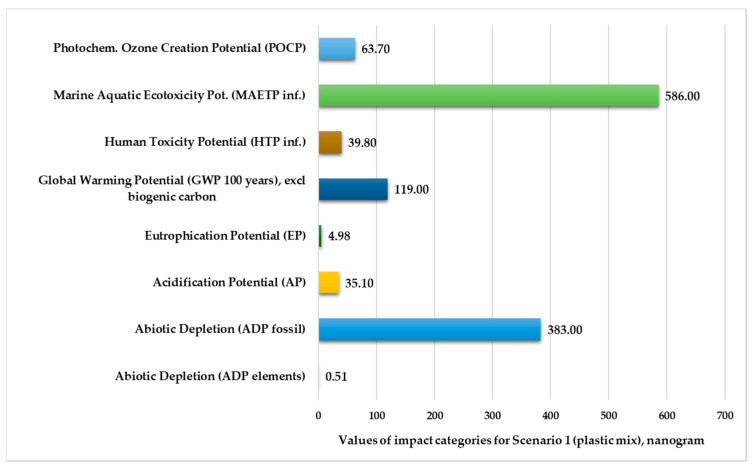
Environmental impact categories for Scenario 1 in the production stage with transport (functional unit: 28 kg plastic mix product).

**Figure 4 polymers-13-00777-f004:**
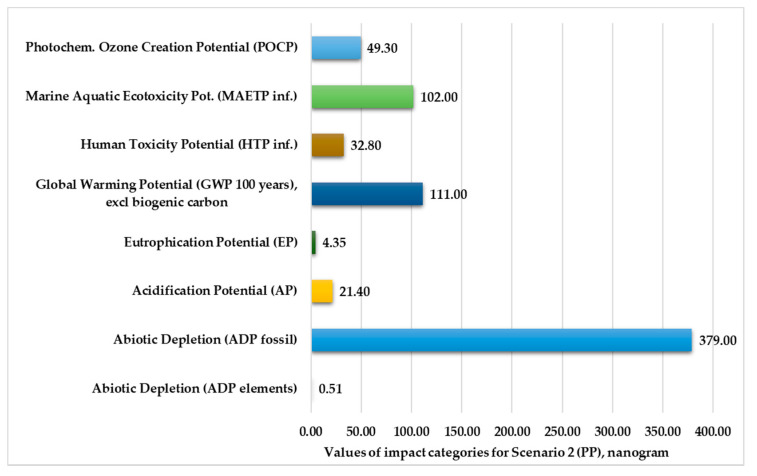
Environmental impact categories for Scenario 2 in the production stage with transport.

**Figure 5 polymers-13-00777-f005:**
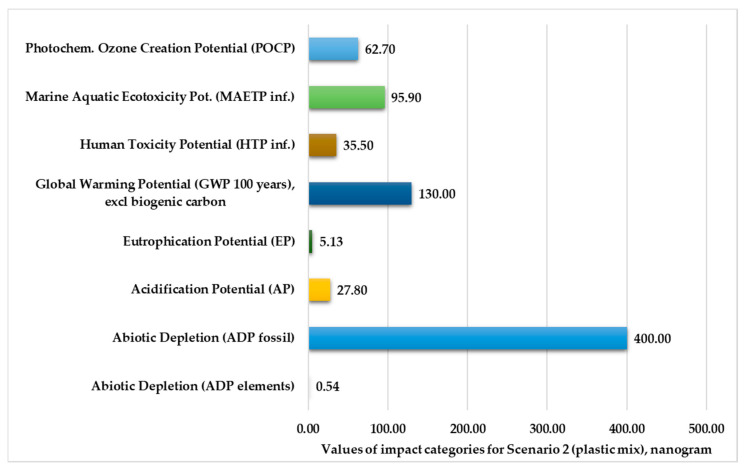
Environmental impact categories for Scenario 2 in the production stage with transport (functional unit: 28 kg mixed-plastic product).

**Figure 6 polymers-13-00777-f006:**
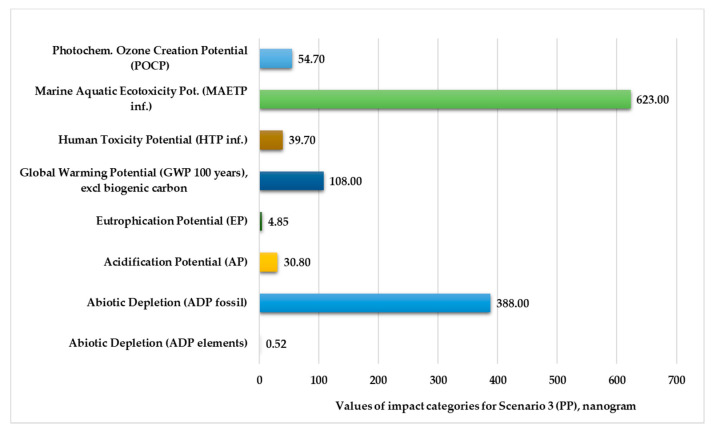
Environmental impact categories for Scenario 3 in the production stage with transport (functional unit: 28 kg polypropylene product. Normalization reference: CML 2016, EU 25+3, year 2000, excl. biogenic carbon. Weighting method: thinkstep LCIA Survey 2012, Europe, CML 2016, excl. biogenic carbon).

**Figure 7 polymers-13-00777-f007:**
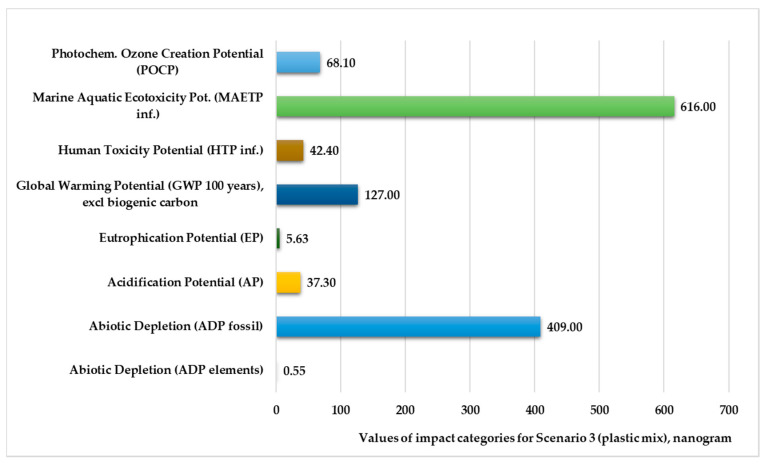
Environmental impact categories for Scenario 3 in the production stage with transport (functional unit: 28 kg mixed-plastic product).

**Figure 8 polymers-13-00777-f008:**
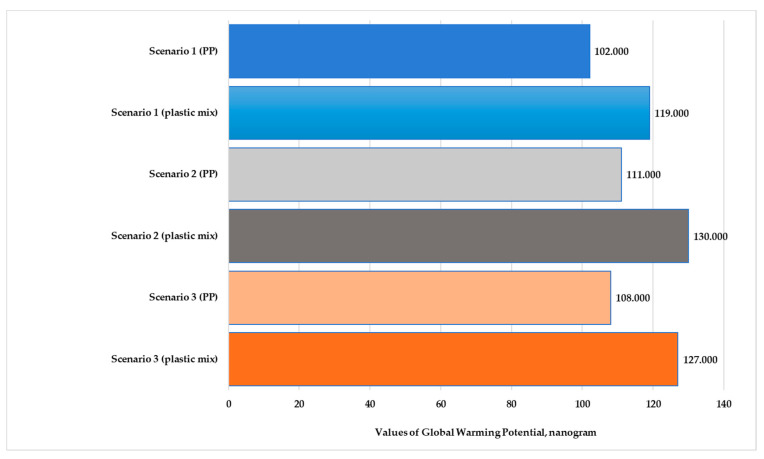
Values of global warming potential for polypropylene and mixed-plastic products for the different scenarios (functional unit: 28 kg product. Normalization reference: CML 2016, EU 25+3, year 2000, excl. biogenic carbon. Weighting method: thinkstep LCIA Survey 2012, Europe, CML 2016, excl. biogenic carbon).

**Figure 9 polymers-13-00777-f009:**
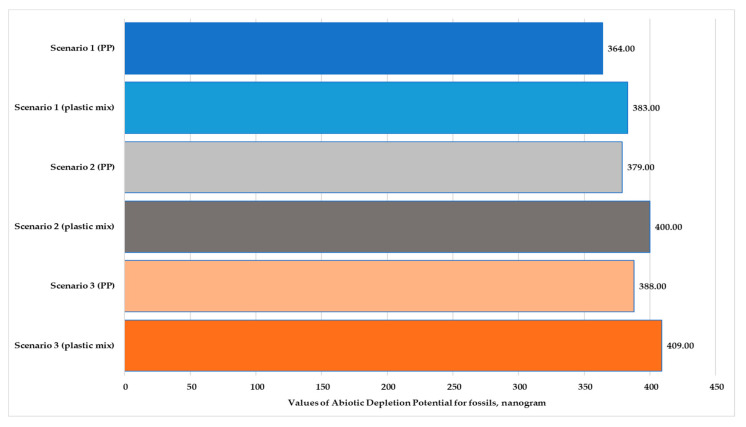
Values of abiotic depletion for fossil fuels of polypropylene and plastic mix products for the different scenarios (functional unit: 28 kg product. Normalization reference: CML 2016, EU 25+3, year 2000, excl. biogenic carbon. Weighting method: thinkstep LCIA Survey 2012, Europe, CML 2016, excl. biogenic carbon).

**Figure 10 polymers-13-00777-f010:**
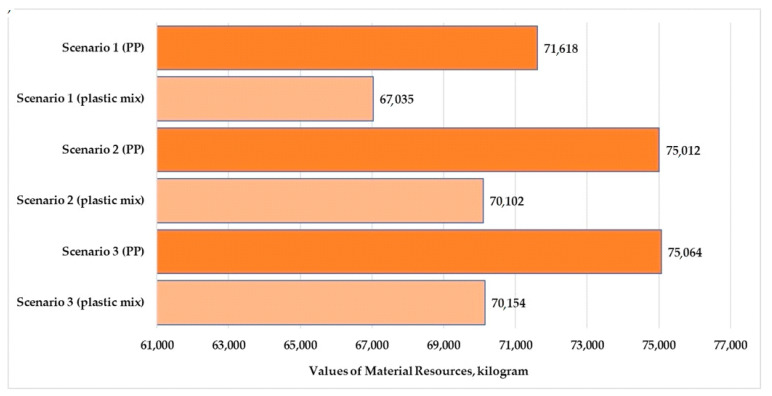
Values of material resources for polypropylene and mixed-plastic products for the different scenarios (functional unit: 28 kg product).

**Figure 11 polymers-13-00777-f011:**
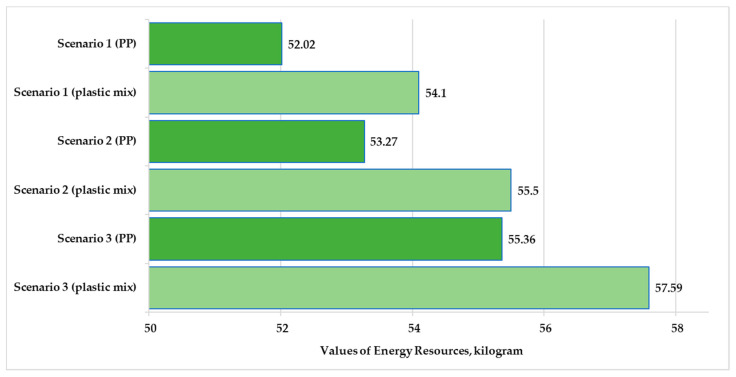
Values of energy resources for polypropylene and mixed-plastic products for the different scenarios (functional unit: 28 kg product).

**Figure 12 polymers-13-00777-f012:**
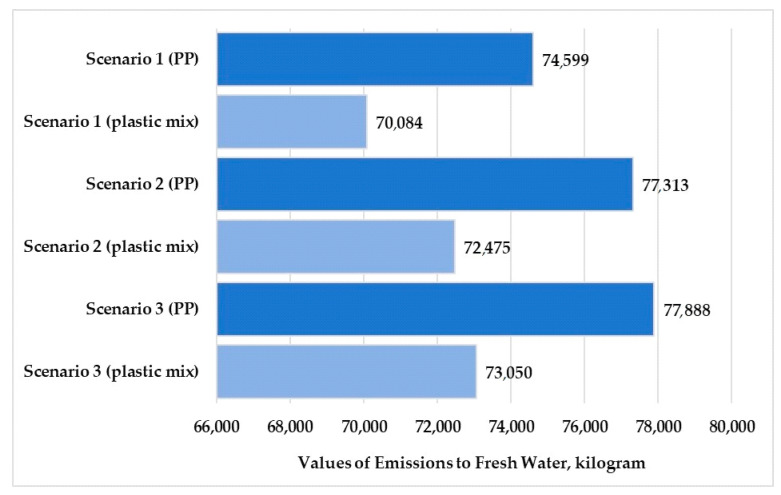
Values of emissions to fresh water for polypropylene and mixed-plastic products for the different scenarios (functional unit: 28 kg product).

**Figure 13 polymers-13-00777-f013:**
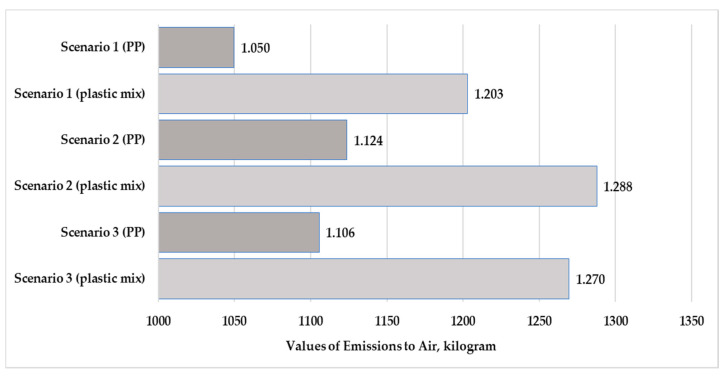
Values of emissions to air for polypropylene and mixed-plastic mix products for the different scenarios (functional unit: 28 kg product).

**Table 1 polymers-13-00777-t001:** The examined environmental impact categories [28,29].

Impact Categories	Equivalent
Abiotic Depletion *ADP elements*, *ADPE*	kg Sb Equivalent
Abiotic Depletion *ADP fossil*, *ADPF*	MJ
Acidification Potential *AP*	kg SO_2_ Equivalent
Eutrophication Potential *EP*	kg Phosphate Equivalent
Freshwater A. Ecot. P. *FAETP inf.*	kg DCB Equivalent
Global Warming Pot. *GWP 100 years*	kg CO_2_ Equivalent
Human Toxicity Potential *HTP inf.*	kg DCB Equivalent
Marine A. Ecotox. Pot. *MAETP inf.*	kg DCB Equivalent
Photochem. Ozone Creat. Pot. *POCP*	kg Ethylene Equivalent
Terrestric Ecotox. Pot. *TETP inf.*	kg DCB Equivalent
Ozone Depletion Pot. *ODP steady state*	kg R11 Equivalent

**Table 2 polymers-13-00777-t002:** Environmental impacts in the production stage of the plastic products for Scenario 1 (functional unit: 1 kg of plastic product. Normalization reference: CML 2016, EU 25+3, year 2000, excluding biogenic carbon. Weighting method: thinkstep LCIA Survey 2012, Europe, CML 2016, excluding biogenic carbon).

Impact Categories	PP (ng)	Plastic Mix (ng)	Difference (%)
Abiotic Depletion *ADP fossil*	364.000	383.000	5.00
Acidification Potential *AP*	29.00	35.10	17.00
Eutrophication Potential *EP*	4.25	4.98	15.00
Global Warming Pot. *GWP 100 years*	102.000	119.000	14.00
Human Toxicity Potential *HTP inf.*	37.20	38.90	4.00
Marine A. Ecotox. Pot. *MAETP inf.*	525.00	586.00	10.00
Photochem. Ozone Creat. Pot. *POCP*	50.50	63.70	21.00
Abiotic Depletion *ADP elements*	0.49	0.51	4.00

**Table 3 polymers-13-00777-t003:** Environmental impacts in the production stage of the plastic products for Scenario 2 (functional unit: 1 kg of plastic product. Normalization reference: CML 2016, EU 25+3, year 2000, excluding biogenic carbon. Weighting method: thinkstep LCIA Survey 2012, Europe, CML 2016, excluding biogenic carbon).

Impact Categories	PP (ng)	Plastic Mix (ng)	Difference (%)
Abiotic Depletion *ADP fossil*	379.000	400.000	5.00
Acidification Potential *AP*	21.40	27.80	23.00
Eutrophication Potential *EP*	4.35	5.13	15.00
Global Warming Pot. *GWP 100 years*	111.000	130.000	15.00
Human Toxicity Potential *HTP inf.*	32.80	35.50	8.00
Marine A. Ecotox. Pot. *MAETP inf.*	102.00	95.90	6.00
Photochem. Ozone Creat. Pot. *POCP*	49.30	62.70	21.00
Abiotic Depletion *ADP elements*	0.51	0.54	6.00

**Table 4 polymers-13-00777-t004:** Environmental impacts in the production stage of the plastic products for Scenario 3 (functional unit: 1 kg of plastic product. Normalization reference: CML 2016, EU 25+3, year 2000, excluding biogenic carbon. Weighting method: thinkstep LCIA Survey 2012, Europe, CML 2016, excluding biogenic carbon).

Impact Categories	PP (ng)	Plastic Mix (ng)	Difference (%)
Abiotic Depletion *ADP fossil*	388.000	409.000	5.00
Acidification Potential *AP*	30.80	37.30	18.00
Eutrophication Potential *EP*	4.85	5.63	14.00
Global Warming Pot. *GWP 100 years*	108.000	127.000	15.00
Human Toxicity Potential *HTP inf.*	39.70	42.40	7.00
Marine A. Ecotox. Pot. *MAETP inf.*	623.000	616.000	2.00
Photochem. Ozone Creat. Pot. *POCP*	54.70	68.10	20.00
Abiotic Depletion *ADP elements*	0.52	0.55	6.00

**Table 5 polymers-13-00777-t005:** Environmental impacts of the polypropylene product for different scenarios (functional unit: 1 kg of polypropylene product. Normalization reference: CML 2016, EU 25+3, year 2000, excluding biogenic carbon. Weighting method: thinkstep LCIA Survey 2012, Europe, CML 2016, excluding biogenic carbon).

Environmental Impacts of PP Product	Scenario 1 (ng)	Scenario 2 (ng)	Scenario 3 (ng)
Abiotic Depletion *ADP fossil*	364.000	379.000	388.000
Acidification Potential *AP*	29.00	21.40	30.80
Eutrophication Potential *EP*	4.25	4.35	4.85
Global Warming Pot. *GWP 100 years*	102.000	111.000	108.000
Human Toxicity Potential *HTP inf.*	37.20	32.80	39.70
Marine A. Ecotox. Pot. *MAETP inf.*	525.00	102.00	623.000
Photochem. Ozone Creat. Pot. *POCP*	50.50	49.30	54.70
Abiotic Depletion *ADP elements*	0.49	0.51	0.52

**Table 6 polymers-13-00777-t006:** Environmental impacts of the mixed-plastic product for different scenarios (functional unit: 1 kg of plastic mix product. Life cycle impact assessment method: CML 2016).

Environmental Impacts of Plastic Mix Product	Scenario 1 (ng)	Scenario 2 (ng)	Scenario 3 (ng)
Abiotic Depletion *ADP fossil*	383.000	400.000	409.000
Acidification Potential *AP*	35.10	27.80	37.30
Eutrophication Potential *EP*	4.98	5.13	5.63
Global Warming Pot. *GWP 100 years*	119.000	130.000	127.000
Human Toxicity Potential *HTP inf.*	38.90	35.50	42.40
Marine A. Ecotox. Pot. *MAETP inf.*	586.00	95.90	616.000
Photochem. Ozone Creat. Pot. *POCP*	63.70	62.70	68.10
Abiotic Depletion *ADP elements*	0.51	0.54	0.55

## Data Availability

Data sharing is not applicable to this article.

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
