# Peer review of "Life Cycle Assessment Model of Plastic Products: Comparing Environmental Impacts for Different Scenarios in the Production Stage"

_polymers, 2021, doi:10.3390/polym13050777_

Round 1

Reviewer 1 Report

Overall Comment

The author presented a lengthy manuscript on the modelling of LCA for plastic products. Overall, each section of the manuscript has weak connection, which needs to be addressed. During the review process, it was also found that the paper showed high similarity in terms of writeup and paper structure as per https://www.mdpi.com/2073-4360/12/9/1901/htm , which is unacceptable. These issues must be addressed accordingly as its current form is not satisfactory to the standard of the journal. Please find my comments as follows:

Title:

The title of the manuscript “Life Cycle Assessment Model of Plastic Products: Comparing Environmental Impacts for Different Scenarios in the Production Stage” seems to be vague. It is suggested that author revised the title of the manuscript as follows:

“Life Cycle Assessment Model of Plastic Products and Mixtures: Evaluation of Environmental Impacts in the Production Stage”

Abstract:

Please revise the whole abstract as this showed direct plagiarism as per https://www.mdpi.com/2073-4360/12/9/1901/htm . The results must also be indicated into the abstract in past tense. Research questions are to be removed, as they are not appropriate in this section. Some numerical data can be included in the abstract as support.

Introduction

If author wants to explain LCA in the introduction, paragraph 1 to 3 can be combined and shortened accordingly, as these paragraphs are wordy and showed no direction. Please define LCA at the start, give example of LCA studies, followed by their advantages. And what are the powerful LCA modelling software available, why GABI is stronger compared to other software?

Paragraph 4-5 needs to be shortened again, and introduce how LCA are used in production, instead of discussing about irrelevant items.

Again combine paragraphs 6 – 8 together, and make the writeup more concise. And please check the in-text citation formats. Overall, the introduction shows no proper structure and needs to be revised greatly to show criticality.

Materials and Methods

Despite this research is modelling work, the chemical composition of the plastics (PP, PE, PET) should be listed in this section as well, as well as their chemical properties. Please write full-form before using abbreviation. Kindly list the scenarios in a list form or table, instead of writeup as this will burden the reader’s to understand the scenarios. The last paragraph in Section 2.1 should be removed from this section

Please create a table to list down the boundary conditions, along with the SI units. What are the specification of the function units?

Section 2.3 and Section 2.4 can be combined. Please shorten Section 2.4. When you said that industry data were used, kindly provide the source. Kindly include the reason for the following statement:

“The amount of energy used to heat, cool, and light the injection moulding space is not included in the system boundaries of this LCI. The following components of each system are not included in this life cycle inventory study: capital equipment, miscellaneous materials, and additives.” Please move Table 1 into this section

Results and Discussion

Section 3.1 discusses the production stage setup. While author provides the values for modelling (i.e: 7 % as maximum product loss; 50 % of water recirculation), justifications should be provided with supporting references.

Section 3.2 presents the results of the modelling outcome. However, due to the enormous graphs involved, it is suggested to combine some of the graphs. Figure 2 and Figure 3 can be combined together. This applies also to Figure 4 and 5; Figure 6 and 7. Furthermore, no comparison in terms of writeup was provided between scenarios. This should not be the case, and their comparisons also indicate whether your models are accurate.

Section 3.3. Table 2 is missing. Similarly the results are being compared without critical discussion. Author must relate the results to the current industry practice, and how this research work can impact thereafter. I recommend that the author rewrite this part completely. Combine figures as necessary to ensure the readability of the manuscript.

Section 3.4. Please make Fig 10 – 13 into one single fig for direct comparison. The values of energy resources are in kg or other unit? Comment as be above

Conclusions

The conclusion is too long, and some of those analysis writeup (paragraph 3 and 4) should be placed into the discussion side. The basis of the study was 28 kg, however in conclusion, 29 kg was written instead. Please summarise the results in just one paragraph, which scenario gives the best outcome and with the numerical data.

Reviewer 2 Report

An interesting lifecycle assessment article on plastics and is in right scope of the journal, it needs some revisions though. My comments are below;

  1. Please improve introduction part, with latest LCA studies on plastic waste and some facts and figures on yearly waste, potential risks too. Moreover, should discuss the strength of LCA tool and use of different lca systems, i-o lca, hybrid lca etc., why you choose this type, for activity or research aim. Add followings to induce flexibility of LCA in various fields. 1) Global consumption of flame retardants and related environmental concerns: A study on possible mechanical recycling of flame retardant textiles. 2) Life cycle assessment of flame retardant cotton textiles with optimized end-of-life phase. 3) Propelling textile waste to ascend the ladder of sustainability: EOL study on probing environmental parity in technical textiles.
  2. I suggest to add detail on how raw material was derived and how it is incorporated in LCI inventory, perhaps the possible limitations of micro plastics, and their waste too.
  3. Adding scenarios is good, however, you should talk why the categories give different values, should talk more on LCIA.
  4. What about sensitivity analysis? The authors should add some comments about it.
  5. Conclusion part should be specific, discussing alternate scenario choice with specific values.